# Deep-Learning-Based Estimation of the Spatial QRS-T Angle from Reduced-Lead ECGs

**DOI:** 10.3390/s22145414

**Published:** 2022-07-20

**Authors:** Ana Santos Rodrigues, Rytis Augustauskas, Mantas Lukoševičius, Pablo Laguna, Vaidotas Marozas

**Affiliations:** 1Biomedical Engineering Institute, Kaunas University of Technology, 51423 Kaunas, Lithuania; vaidotas.marozas@ktu.lt; 2Department of Automation, Kaunas University of Technology, 51367 Kaunas, Lithuania; rytis.augustauskas@ktu.lt; 3Faculty of Informatics, Kaunas University of Technology, 51368 Kaunas, Lithuania; mantas.lukosevicius@ktu.lt; 4Biomedical Signal Interpretation and Computational Simulation (BSICoS) Group, Aragón Institute of Engineering Research (I3A), IIS Aragón, University of Zaragoza, 50018 Zaragoza, Spain; laguna@unizar.es; 5Biomedical Research Networking Center (CIBER), 50018 Zaragoza, Spain; 6Faculty of Electrical and Electronics Engineering, Kaunas University of Technology, 51367 Kaunas, Lithuania

**Keywords:** wearable devices, consumer healthcare devices, cardiovascular heath assessment, unobtrusive monitoring, machine learning, regression, composite loss function

## Abstract

The spatial QRS-T angle is a promising health indicator for risk stratification of sudden cardiac death (SCD). Thus far, the angle is estimated solely from 12-lead electrocardiogram (ECG) systems uncomfortable for ambulatory monitoring. Methods to estimate QRS-T angles from reduced-lead ECGs registered with consumer healthcare devices would, therefore, facilitate ambulatory monitoring. (1) Objective: Develop a method to estimate spatial QRS-T angles from reduced-lead ECGs. (2) Approach: We designed a deep learning model to locate the QRS and T wave vectors necessary for computing the QRS-T angle. We implemented an original loss function to guide the model in the 3D space to search for each vector’s coordinates. A gradual reduction of ECG leads from the largest publicly available dataset of clinical 12-lead ECG recordings (*PTB-XL*) is used for training and validation. (3) Results: The spatial QRS-T angle can be estimated from leads {*I*, *II*, *aVF*, *V2*} with sufficient accuracy (absolute mean and median errors of 11.4° and 7.3°) for detecting abnormal angles without sacrificing patient comfortability. (4) Significance: Our model could enable ambulatory monitoring of spatial QRS-T angles using patch- or textile-based ECG devices. Populations at risk of SCD, like chronic cardiac and kidney disease patients, might benefit from this technology.

## 1. Introduction

Despite recent advances in treating cardiovascular diseases, sudden cardiac death (SCD) remains the leading cause of mortality, accounting for approximately 20% of all deaths in western societies [1,2]. Dangerous arrhythmias precipitated by abnormalities in ventricular repolarization often precede SCD [3,4,5]. Various markers of abnormal repolarization in the electrocardiogram (ECG) have been proposed to stratify the risk of SCD, including changes in ST-segment [6] and QT interval lengthening [7]. However, those that evaluate the similarity between the direction of depolarization and repolarization, such as the spatial QRS-T angle, are deemed the most promising [8,9,10]. Unfortunately, the estimation of QRS-T angle is mainly restricted to clinical settings. The conventional approach for the spatial QRS-T angle estimation [8,11] is uncomfortable for patients as it requires a standard 12-lead ECG, hindering the possibility of harnessing the diagnostic value of the QRS-T angle for out-of-hospital early detection of dangerous cardiac events. Methods to estimate the spatial QRS-T angle from a set of reduced-lead ECGs would, therefore, be of clinical importance for ambulatory monitoring. Such methods could be deployed in consumer healthcare devices and facilitate out-of-hospital monitoring of the QRS-T angle in populations at risk of life-threatening cardiac events.

Thus far, the spatial QRS-T angle is estimated exclusively from orthogonal signals, either the vectorcardiogram (VCG) [8] or orthogonalized 12-lead ECGs [11], that depict the electrical activity of the heart in the *XYZ* plane. The spatial QRS-T angle denotes the angle between the QRS- and T-wave vectors in the 3D space. In the absence of Frank’s lead system, the VCG is regularly reconstructed from the standard 12-lead ECG by applying one of the various mathematical transformations that convert 12-lead ECGs into a set of three orthogonal leads [12,13,14]. Registration of a 12-lead ECG, or even Frank’s VCG, requires the patient to use eight or ten electrodes [13], causing considerable discomfort. Configuring eight-to-ten electrodes as specified in clinical Holter monitors is usually an intricate task for the ordinary patient, making it unfeasible even to request patients to set up such devices for intermittent monitoring of the QRS-T angle. Conversely, consumer healthcare devices, designed to ameliorate patient discomfort, are compact, practical, and easy to configure. However, the number of ECG leads registered by consumer healthcare devices is limited to a few frontal with one-to-two precordial leads. These sets of leads are insufficient to reconstruct the VCG, thus precluding the employment of any of the existing methods for spatial QRS-T angle estimation.

Deep neural networks have demonstrated tremendous capabilities to extract key data insights from sets of reduced-lead ECGs instead of the standard 12-leads [15]. For instance, 1D convolutional neural networks (CNNs) have been shown to detect arrhythmias in clinical [16] and ambulatory [17,18] single-lead ECGs, and even sleep apnea [19,20] with up to 97.1% accuracy [20]. CNNs have also reconstructed the standard 12-lead ECG from a few measured leads [21,22]. The ostensible potential of CNNs motivated us to investigate whether it is possible to estimate the spatial QRS-T angle using a set of reduced-lead ECGs. We hypothesize that, by using 12-lead ECGs, a model can be trained to predict the VCG-derived QRS and T vectors from a specific subset of ECG leads.

We present a 1D convolutional neural network (CNN1D) to estimate the spatial QRS-T angle from signal-averaged heartbeats of reduced-lead ECGs. Since the spatial location of QRS and T vectors is largely dependent on the cardiac conduction axis, the model is designed to return the coordinates of both vectors as output. Our study introduces an original composite loss function that uses the QRS-T angle and the Euclidean distance between the vectors to guide the model throughout the 3D space. The model is developed and validated on the *PTB-XL* [23] dataset, the current largest publicly available database of clinical 12-lead ECG recordings. We investigate the performance of our model in sets of ECG leads that can conveniently be recorded with patch-based consumer healthcare devices. Lastly, we explore the feasibility of measuring the spatial QRS-T angle from solely frontal leads, aiming to understand the future challenges of deep-learning-based QRS-T angle estimation for ambulatory monitoring. To our knowledge, this is the first study to examine the feasibility of estimating the QRS-T angle from reduced-lead ECGs.

This article is organized as follows. Section 2 and Section 3 describe the conventional and the proposed deep-learning-based approaches for the spatial QRS-T angle estimation. Section 4 discloses information about the training and validation datasets, including the data preparation and labeling procedures. Section 5 defines the investigative methodology and performance evaluation. Finally, Section 6 presents the results, followed by a discussion and conclusions in Section 7 and Section 8.

## 2. Conventional Approach for QRS-T Angle Estimation

The spatial QRS-T angle is estimated from a set of three orthogonal leads, obtained either by applying orthogonalization methods to 12-lead ECGs [11,24] or, conventionally, the VCG. The VCG, composed of leads *XYZ*, reflects the electrical activity of the heart in the orthogonal planes [25]: frontal (*XY*), transverse (*XZ*), and sagittal (*YZ*). In essence, the VCG depicts heartbeats as a trajectory of leads *XYZ* over time,
(1)$$\vec{v}\left(t\right)=\left[x(t),y(t),z(t)\right],$$
in which the depolarization (QRS) and repolarization (T) phases of a heartbeat are represented as two loops: (2)$${\vec{v}}_{QRS}\left(t\right)=\vec{v}\left(t\right)-{\vec{v}}_{0},\;\mathrm{with}\;t\in \left\{t_{QRS_{o}},\ldots ,t_{QRS_{e}}\right\},$$
(3)$${\vec{v}}_{T}\left(t\right)=\vec{v}\left(t\right)-{\vec{v}}_{0},\;\mathrm{with}\;t\in \left\{t_{T_{o}},\ldots ,t_{T_{e}}\right\},$$
where $t_{QRS_{o}}$, $t_{T_{o}}$, $t_{QRS_{e}}$, and $t_{T_{e}}$ are the onset and offset of QRS and T loops respectively. Following the guidelines in [26], the origin of both loops ${\vec{v}}_{0}$ is estimated as: (4)$${\vec{v}}_{0}=\underset{t}{\mathrm{median}}\left(\vec{v}(t)\right),\;\mathrm{where}\;t\in \left\{t_{QRS_{o}}-\tau_{0},\ldots ,t_{QRS_{o}}\right\}\;\mathrm{and}\;\tau_{0}=25ms.$$
Since inaccuracies in heartbeat delineation can generate significant errors in the estimation of QRS-T angle, the onsets $t_{QRS_{o}}$, $t_{T_{o}}$, and offsets $t_{QRS_{e}}$, $t_{T_{e}}$ are adjusted as instructed in [26].

The spatial QRS-T angle measures the dissimilarity between the orientation of the QRS and T loops in the *XYZ* space and is calculated as: (5)$$\alpha =\arctan\left(\frac{∥{\vec{u}}_{QRS}\times {\vec{u}}_{t}∥}{{\vec{u}}_{QRS}\;\cdot \;{\vec{u}}_{T}}\right),$$
where ${\vec{u}}_{QRS}$ and ${\vec{u}}_{T}$ are vectors that depict the dominant orientation of QRS and T loops respectively. The loop orientation is most commonly defined in the time instance $t=t_{max}$ where the maximum magnitude [8] of ${\vec{v}}_{QRS}\left(t\right)$ or ${\vec{v}}_{T}\left(t\right)$ is verified: (6)$${\vec{u}}_{QRS}={\vec{v}}_{QRS}\left(t_{QRS_{max}}\right),\;\mathrm{where}\;t_{QRS_{max}}=\underset{t}{arg max}\left(∥{\vec{v}}_{QRS}\left(t\right)∥\right),$$
(7)$${\vec{u}}_{T}={\vec{v}}_{T}\left(t_{T_{max}}\right),\;\mathrm{where}\;t_{T_{max}}=\underset{t}{arg max}\left(∥{\vec{v}}_{T}\left(t\right)∥\right).$$

Although intuitive, defining the loop spatial orientation as the vector having the maximal magnitude at a single-time instance is an oversimplification, as it assumes that the morphology of the QRS and T loops is unambiguous enough to have a well-defined spatial orientation. In abnormal ECGs, the spatial orientation of the loops, in particular the QRS loop, is too complex to be represented by a vector in a single instance in time. In fact, estimation of the QRS-T angle using ${\vec{v}}_{QRS}\left(t_{QRS_{max}}\right)$ and ${\vec{v}}_{T}\left(t_{T_{max}}\right)$ has been associated with higher errors and poorer reproducibility [27], namely in unhealthy ECGs.

One strategy to tackle the problem of defining the underlying spatial orientation of the QRS loop is the *total cosine R-to-T* (TCRT) [11] method. TCRT defines the QRS-T angle as the average cosine of all angles between ${\vec{v}}_{T}\left(t_{T_{max}}\right)$ and every vector within the QRS loop that exceed 70% of the maximum vector magnitude ${\vec{v}}_{QRS}\left(t_{QRS_{max}}\right)$ [28]. However, computation of an averaged angle can become problematic in sets of reduced-lead ECGs that do not carry the same amount of spatial information as the VCG (see Section 3.2). Consequently, we adopt a strategy similar to TCRT, but instead of deriving the average cosine, we define ${\vec{u}}_{QRS}$ and ${\vec{u}}_{T}$ as the average of all vectors exceeding 70% of the maximum vector magnitude within the corresponding loops: (8)$${\vec{u}}_{QRS}=\underset{t}{\mathrm{mean}}\left({\vec{v}}_{QRS}\left(t\right)\right),\;\mathrm{where}\;t\in \left\{t\;|\;∥{\vec{v}}_{QRS}\left(t\right)∥\geq 0.7\;∥{\vec{v}}_{QRS}\left(t_{QRS_{max}}\right)∥\right\},$$
(9)$${\vec{u}}_{T}=\underset{t}{\mathrm{mean}}\left({\vec{v}}_{T}\left(t\right)\right),\;\mathrm{where}\;t\in \left\{t\;|\;∥{\vec{v}}_{T}\left(t\right)∥\geq 0.7\;∥{\vec{v}}_{T}\left(t_{T_{max}}\right)∥\right\}.$$
The spatial QRS-T angle is then calculated as the angle between ${\vec{u}}_{QRS}$ and ${\vec{u}}_{T}$ defined by Equations (Equation 8) and (Equation 9).

## 3. Deep-Learning-Based Approach for QRS-T Angle Estimation

We propose a deep learning model to estimate the spatial QRS-T angle using a set of reduced-lead ECG. The model takes the signal-averaged beats from a set of leads as an input, and produces the three coordinates of ${\vec{u}}_{QRS}$ and ${\vec{u}}_{T}$, i.e., ${\vec{u}}_{QRS}=\left(x_{QRS},\;y_{QRS},\;z_{QRS}\right)$ and ${\vec{u}}_{T}=\left(x_{T},\;y_{T},\;z_{T}\right)$, as the output. The set of input leads varies in different experiments, as discussed in Section 5.1.

Using 12-lead ECGs, we can compute the reference (target) VCG vectors ${\vec{u}}_{QRS}$ and ${\vec{u}}_{T}$ using the conventional approach described in Section 2, and train the model to produce the estimates ${\hat{\vec{u}}}_{QRS}$ and ${\hat{\vec{u}}}_{T}$ of the targets from specific subsets of ECG leads. The estimated QRS-T angle can then be calculated as the angle between the estimated vectors, ${\hat{\vec{u}}}_{QRS}$ and ${\hat{\vec{u}}}_{T}$, using Equation (Equation 5). The model is purposely designed to produce the vectors instead of the angle directly to harness the available spatial information when training the model (see Section 3.2). Figure 1 presents an overview of our deep-learning-based approach.

From this point onwards, the circumflex symbol denotes variables estimated by the model: ${\hat{\vec{u}}}_{QRS}$, ${\hat{\vec{u}}}_{T}$, and the QRS-T angle $\hat{\alpha }$ between them; whereas ${\vec{u}}_{QRS}$ and ${\vec{u}}_{T}$ are the VCG target vectors and α is the angle between them.

### 3.1. Deep Learning Model Architecture

A 1D convolutional neural network (CNN1D) with a regression output is the baseline architecture for our proposed deep learning model. The model is trained end-to-end using error backpropagation and gradient descent. It can conceptually be divided into two main parts: the feature extraction and the regression networks.

Since distinct subsets of ECG leads may entail different configurations, we first describe the baseline architecture of our model, and then detail hyperparameter tuning.

#### 3.1.1. Feature Extraction Network

The feature extraction network is composed of *D* blocks of layers connected sequentially. Each block consists of two “layer structures”, except the first block, which only includes one. Each layer structure is a sequence of: a full 1D convolutional layer with *k* feature kernels of size 3×1 and a stride of 1, followed by a layer normalization and an activation function (Figure 2b). The layer normalization balances the intermediate features to have a mean close to 0.0 and a standard deviation close to 1.0 using scale and shift parameters that are trainable for each feature map. Leaky Rectified Linear Unit (Leaky ReLU) with the negative slope coefficient of 0.1 is the chosen activation function.

In the first block, a depthwise convolutional layer is employed instead of a full convolution (Figure 2a). A depthwise convolution allows the model to learn lead-specific features separately, as each lead can carry relevant information on the position of each coordinate of ${\vec{u}}_{QRS}$ and ${\vec{u}}_{T}$. Because depthwise convolution layers generate feature maps for each individual lead, the initial number of kernels *k* is distributed across all leads, giving $\frac{k}{j-1}$ each, where *j* is the number of input leads. This avoids having a larger feature map in the first layer than in the second.

Residual connections (Figure 2c) are introduced from the second block d=2 to the block number d=D−1 to maintain data flow throughout the network and avoid gradient degradation during training. Prior to addition, 1×1 convolution is used on the residual connection to equalize the number of feature maps between the layers. The number of filters increases by a factor of 2 in every subsequent residual block. Abstraction of the most significant features is performed with max pooling at the end of blocks d= [2: D−1], whereas global average pooling is implemented to finalize the last block d=D of the feature extraction network. To avoid overfitting, *dropout* with a probability of 0.25 is applied after feature extraction.

#### 3.1.2. Regression Network

The resultant feature map is connected to the fully-connected layers (regression), which learns to associate the abstracted features with the six neurons in its output: one for each of the three coordinates of ${\hat{\vec{u}}}_{QRS}=\left({\hat{x}}_{QRS},\;{\hat{y}}_{QRS},\;{\hat{z}}_{QRS}\right)$ and ${\hat{\vec{u}}}_{T}=\left({\hat{x}}_{T},\;{\hat{y}}_{T},\;{\hat{z}}_{T}\right)$. The regression network consists of three dense layers, the first two followed by layer normalization and Leaky ReLU activation function. The output layer consisting of six neurons is followed by linear activation. Since ECGs can exhibit sex- and age-related dissimilarities in morphology [29] that can affect the QRS-T angle [30,31], metadata about sex (0 for males, or 1 for females) and age (scaled from 0.0 to 1.0) are concatenated to the first layer in the regressive model part. Providing the hints to the model about a possible association between ECGs and the metadata may be valuable when the available spatial information in the input leads is reduced.

### 3.2. Loss Function

Since the end goal is to determine the QRS-T angle, the most straightforward approach would be to train the model to estimate the VCG-derived α directly instead of ${\vec{u}}_{QRS}$ and ${\vec{u}}_{T}$, optimizing it with the mean absolute error loss between the α and the estimated $\hat{\alpha }$:(10)$$L_{\alpha }\left(\alpha ,\hat{\alpha }\right)=\frac{1}{n}\;\sum_{i=1}^{n}\left(\left|\alpha_{i}-{\hat{\alpha }}_{i}\right|\right),$$
where $0{}^{\circ}\leq L_{\alpha }\leq 180{}^{\circ}$ and *n* is the batch size.

Direct estimation of the QRS-T angle, albeit intuitive and straightforward, overlooks crucial information about the spatial orientation and position of the QRS and T loops, trivializing the problem of QRS-T angle estimation as explained in Section 2. In sets of reduced-lead ECGs that only carry fragments of all spatial information contained in the VCG, this approach can produce errors in ECGs with visible differences in morphology but similar QRS-T angles. Morphologically different ECGs with QRS-T angles of equivalent range can occur in patients in which the electrical activity of the heart is not conducted in the same direction, that is, the cardiac conduction axis is nonidentical. In two patients with distinct cardiac conduction axes but similar QRS-T angles, the corresponding vectors ${\vec{u}}_{QRS}$ and ${\vec{u}}_{T}$ of each patient are located in different planes (octants) in the 3D space, but the angle between them is still alike (Figure 3a).

To address these scenarios, we devise the model to locate the coordinates of ${\vec{u}}_{QRS}$ and ${\vec{u}}_{T}$ instead of α directly, allowing the model to harness any spatial information available in the input leads. The model is guided throughout the 3D space using the Euclidean distance as the parameter to be minimized in the backpropagation algorithm. The 3D Euclidean distance (d) between the coordinates of $\vec{u}$ and $\hat{\vec{u}}$ is computed as: (11)$$L_{d}\left(\vec{u},\hat{\vec{u}}\right)=\frac{1}{n}\;\sum_{i=1}^{n}\sqrt{{\left(x_{i}-\hat{x_{i}}\right)}^{2}+{\left(y_{i}-\hat{y_{i}}\right)}^{2}+{\left(z_{i}-\hat{z_{i}}\right)}^{2}},$$
where, $0\leq L_{d}\leq 2$ if $\vec{u}$ and $\hat{\vec{u}}$ have a magnitude of 1 (i.e., unit vectors). Two unit vectors $\vec{a}$ and $\vec{b}$ with opposite directions are circumscribed by an angle of 180°, thus translating into an Euclidean distance equal to the sum of their magnitudes: $∥\vec{a}∥+∥\vec{b}∥=1+1=2$.

In order for $\hat{\alpha }$ to be equal to α, only the direction, but not the magnitude, of the estimated $\hat{\vec{u}}$ has to match the target $\vec{u}$. Given that the Euclidean distance between two vectors also accounts for differences in magnitude, which is undesirable in this case, we transform $\vec{u}$ and $\hat{\vec{u}}$ to unit vectors prior to calculating $L_{d}$. Calculating the Euclidean distance between unit vectors avoids wrongfully calculating a high loss in cases of two vectors with the same direction but discrepant magnitudes, which should be zero in this application. The principle is similar to the *cosine similarity*. However, the Euclidean distance is preferable for this case scenario as it permits to navigate throughout each axis in *XYZ* plane, whereas the *cosine similarity* only discerns one axis (in the 2D space, the cosine can distinguish quadrant I from II, or IV from III, but not I from IV nor II from III).

Another problem left to address during the training process is cases in which one of the vectors is less complicated to determine than the other (Figure 3b), i.e., the model properly locates one vector but not the other (e.g., $L_{d}\left({\vec{u}}_{T},{\hat{\vec{u}}}_{T}\right)≊$ 0 and $L_{d}\left({\vec{u}}_{QRS},{\hat{\vec{u}}}_{QRS}\right)≊$ 1.2). Significant errors in estimating one vector will inherently affect the accuracy of the QRS-T angle. Since the angle between ${\hat{\vec{u}}}_{QRS}$ and ${\hat{\vec{u}}}_{T}$ needs to be equivalent to α, we mitigate such cases by confining the model’s search grid to preserve the angle $\hat{\alpha }$ between ${\hat{\vec{u}}}_{QRS}$ and ${\hat{\vec{u}}}_{T}$ as close as possible to α. Thus, we define the overall loss as a composite function of (Equation 10) and (Equation 11): (12)$$L=w_{1}\left(L_{d}\left({\vec{u}}_{QRS},{\hat{\vec{u}}}_{QRS}\right)+L_{d}\left({\vec{u}}_{T},{\hat{\vec{u}}}_{T}\right)\right)+w_{2}\;L_{\alpha }\left(\alpha ,\hat{\alpha }\right),$$
where $w_{1}$ and $w_{2}$ are hyperparameters that weigh the penalization factor of $L_{d}$ and $L_{\alpha }$. The proposed composite loss function safeguards the overall accuracy of the model by avoiding that $L_{d}$ of one vector is substantially higher than $L_{d}$ of the other, with the tradeoff of allowing minor errors in the location of both vectors (i.e., $L_{d}≊$ 0.1 instead of $L_{d}≊$ 0), as long as the angle $\hat{\alpha }$ between them is close to α (see Figure 3c). To equalize the scales of $L_{d}$ and $L_{\alpha }$, $L_{\alpha }\left(\alpha ,\hat{\alpha }\right)$ is estimated in radians rather than degrees.

### 3.3. Tuning of Hyperparameters

Several experiments are conducted to find the best architecture for each of the tested subsets of leads according to the hyperparameters $w_{1}$ and $w_{2}$, depth *D*, and the initial number of kernels *k*. The hyperparameters are chosen among the following options: *D* = {2, 3, 4, 5}, *k* = {8, 16}, $w_{1}$ = {0.5, 0.8, 1.0, 1.2, 1.5} ∧ $w_{2}$ = $|1-w_{1}|$, and $w_{2}$ = {0.8, 1.0, 1.2, 1.5} ∧ $w_{1}$ = $|1-w_{2}|$. The hyperparameters *D* and *k* are constrained to the above values due to the following. First, complex CNNs employed for image-based applications are likely an overengineered solution for our problem. Second, smaller CNN architectures enhanced with residual connections and case-specific loss functions can outperform architectures based on regular convolutional blocks [32,33]. Third, lightweight and low-complexity models are preferable for deployment in devices with hardware and computational constraints, such as consumer healthcare devices. Training is performed with a batch size of *n* = 8 at an initial learning rate of 0.001 for 100 epochs. After every 20 epochs, the learning rate is reduced by half.

## 4. Data

The deep learning model is developed and validated on the *Physionet* [34] *PTB-XL* dataset [23], the current largest publicly available dataset of 12-lead ECG recordings. The *PTB-XL* comprises 21,837 clinical recordings of 10 s long ECGs, upsampled to 500 Hz, from 18,885 patients (48% females) with ages ranging from 0 to 95 years. Information on the diagnosis, form, rhythm, and signal quality is provided for all recordings. As to diagnosis, the ECGs are categorized into five different superclasses: Normal (*NORM*), Myocardial Infarction (*MI*), Conduction Disturbance (*CD*), ST/T change (*STTC*), and Hypertrophy (*HYP*). The superclasses are branched into several subclasses, apart from *NORM*.

### 4.1. Data Preparation and Labeling

Leads X, Y, and Z (VCG) are derived from raw ECGs by applying the Kors regression matrix [12], the mathematical transformation that more accurately reconstructs Frank’s VCG from an ECG [13]. The generated 15-lead signals undergo preprocessing comprised of filtering, signal quality assessment, and beat averaging. The target vectors ${\vec{u}}_{QRS}$ and ${\vec{u}}_{T}$ are finally computed from the generated signal-averaged VCG leads to label the data. The code implementation for data labeling together with information regarding the training and validation sets can be found in our GitHub repository in [35]. Figure 4 illustrates the data preparation process.

#### 4.1.1. Signal Preprocessing

*Filtering.* High-frequency noise and baseline wandering are filtered with zero-phase low- and high-pass Butterworth filters with cut-off frequencies of 45 Hz and 0.5 Hz.

*Signal quality assessment.* The signal quality index (SQI) criteria proposed in [36] is applied to each lead individually to eliminate beats of dissimilar morphology, such as ectopic beats or those corrupted by noise. Recordings with at least one lead that contains more than 50% poor-quality beats within the 10 s ECG are considered unanalyzable and hence discarded. ECGs with discernible rhythm disturbances, such as atrial or ventricular flutter or fibrillation, are also excluded from the analysis given their greater predisposition to PQRST delineation errors that can affect the reliability of ${\vec{u}}_{QRS}$ and ${\vec{u}}_{T}$ [26]. Annotations regarding rhythm are provided in the *PTB-XL* dataset. In case of rhythm disturbances like bradycardia, tachycardia or sinus arrhythmia, PQRST delineation can be less problematic when signals are of high-quality; thus, such ECGs are still considered for analysis if 70% of all beats satisfy the SQI criteria.

*Beat averaging.* High-quality beats are aligned using the R-peak as the reference point and averaged, resulting in a signal-averaged heartbeat representative of each chosen lead.

#### 4.1.2. Data Labeling

Our *training labels*, i.e., the target VCG vectors ${\vec{u}}_{QRS}$ and ${\vec{u}}_{T}$ are computed from the three averaged beats of leads *XYZ* using the conventional approach described in Section 2. The QRS and T loops onset and offset, $t_{QRS_{o}}$, $t_{T_{o}}$, $t_{QRS_{e}}$, and $t_{T_{e}}$, and R-peaks are identified with the multilead PQRST delineation algorithm available in the *ECGDeli* [37] toolbox. The onset and offset of the loops are adjusted as instructed in [26]. Note that robust PQRST delineation algorithms are critical to compute reliable *training labels* for developing the model, but are not necessary in future applications in which only averaged heartbeats and metadata are required as input.

Lastly, the averaged beats are downsampled to 250 Hz and zero-padded to 550 samples to equalize their length, as deep learning models require inputs of identical size. Since the standard clinical ECG bandwidth is 0.05 Hz to 100 Hz [38], downsampling the average beats to 250 Hz reduces the computational complexity of the model and the necessary resources (e.g., RAM) for training without compromising crucial signal information. Patient metadata is also added to the *training labels*: information about sex is specified as 0 for males and 1 for females, and age is scaled from 0.0 (0 years) to 1.0 (100 years).

Of 21,837 clinical recordings, 18,618 are eligible for labeling and analysis. In addition to poor-quality ECGs or with complicated rhythm disturbances, we exclude recordings in which the assigned subclass is underrepresented in the dataset, having less than 100 recordings that meet the described SQI criteria. ECGs of rare subclasses have such unusual morphologies that errors can be introduced into the model due to the scarcity of recordings.

### 4.2. Exploratory Data Analysis

Exploratory data analysis is performed on the labeled recordings before splitting the data between the training and validation sets. The goal is to eliminate any statistical bias by ensuring that both sets preserve the same distribution of sex, morphological classes, and the spatial QRS-T angle in the ranges of α = [0:5:180]°, as in the original dataset. We center our exploratory data analysis and subsequent splitting around these three attributes due to the following:Sex-related morphological differences in the ECG may influence the decision of the regression network (see Section 3.1.2); thus, the training set must be proportioned in terms of sex.Each of the morphological classes is characterized by distinctive morphological traits. Since contrastive ECG morphologies can still exhibit QRS-T angles of comparable range, the training set must include a diversity of morphologies to prevent the model from associating a specific range of QRS-T angles with just one subset of particular morphological traits.Randomly splitting the data without considering the uneven distribution of α within specific ranges could result in a disproportionate depiction of specific ranges in the training set, leading to higher errors in other ranges.

Recordings are divided into six morphological classes: the same five diagnostic superclasses stipulated in the *PTB-XL* dataset, *NORM*, *MI*, *CD*, *STTC*, *HYP*, and low magnitude T waves (*LOWM*). A recording is deemed *LOWM* if the ratio between $∥{\vec{u}}_{T}∥$ and $∥{\vec{u}}_{QRS}∥$ < 0.1. Although signals with low magnitude T waves seem to have a higher propensity to QRS-T angle errors [26] and are often discarded [26,39], we consider to be reasonable to incorporate such signals into this study, given that low magnitude T waves are found routinely in clinical practice.

Figure 5 shows the distribution of α across the ranges of α = [0:5:180]°, according to sex and morphological class. The dataset has a median of 52.9° (interquartile range of 63.3°). The distribution of α, albeit balanced between males and females, varies considerably for each morphological class. Although spatial QRS-T angles 15° ≤α≤ 90° comprise the vast majority of the eligible recordings, all other ranges of α are represented by at least 100 recordings, which may be sufficient for deep-learning-based estimation of QRS-T angle with an acceptable error.

### 4.3. Training and Validation Sets

The data is split separately for females and males in each morphological class to ensure an appropriate data allocation between the training and validation sets. The split is performed as follows. For any given morphological class, 80% female ECGs and 80% of male ECGs with α = [*i*:i+5], for every *i* = [0:5:175]°, are randomly assigned to the training set. Given the propensity of *LOWM* signals to display larger errors of α, the 50:50 partition ratio is used for this class instead of 80:20. A smaller partition of the *LOWM* class still enables the class to be adequately represented in the training set without excessively misleading the deep learning model. Figure 6 shows that both the training and validation sets preserve the original distribution of α.

## 5. Experiments and Performance Evaluation

The model is written in Python (v3.8.10) using the Keras abstraction layer on Tensorflow 2.8.0 backend. Training and validation are performed on a desktop computer under Windows 10 environment composed of: Intel^®^ Core^®^ i7-8700k 3.70 GHz CPU with six cores (12-threads), 32 GB of RAM, and NVIDIA^®^ GeForce^®^ GTX 1080Ti.

### 5.1. Selection of Subsets of ECG Leads

We investigate the performance of our model to estimate the spatial QRS-T angle from various subsets of ECGs leads. The goal is to identify how many leads suffice to estimate the QRS-T angle with acceptable accuracy without sacrificing patient comfortability. We start by configuring the baseline architecture of our model using the leads that contain all the 3D spatial information, *XYZ*, from which the target ${\vec{u}}_{QRS}$ and ${\vec{u}}_{T}$ are derived. Next, we progressively trim the number of precordial leads that carry insights about the spatial position of ${\vec{u}}_{QRS}$ and ${\vec{u}}_{T}$ in each of the *X*, *Y*, and *Z* axes. The baseline model architecture is optimized for sets of reduced-lead ECGs that incorporate a minimum of one lead shown to reflect each orthogonal axis: *X* ⊆ {*I*, *V5*, *V6*}; *Y* ⊆ {*II*, *III*, *aVF*}; and *Z* ⊆ {*V1*, *V2*, *V3*} [14].

Since this research ultimately aims to develop a method to facilitate QRS-T angle monitoring in free-living conditions, we only test sets of reduced-lead ECGs that can be acquired from commercialized consumer healthcare devices. Registration of frontal leads is straightforward: all six frontal leads ({*I*, *II*, *III*, *aVL*, *aVR*, *aVF*}) can be derived from any device with two-frontal channels. However, most consumer healthcare devices equipped for frontal and precordial lead registration offer no more than two precordial leads: *V2* and *V6*. Thus, we limit our experiments to the subsets of leads *S* ⊆ {*I*, *II*, *III*, *aVL*, *aVR*, *aVF*, *V2*, *V6*}.

While a decline in performance is anticipated as the number of precordial leads decreases, we also explore as a proof-of-concept the ability of our model to estimate the spatial QRS-T angle from subsets of exclusively frontal leads.

In this article, we only present the results of the best subset of leads: first X, Y, and Z, then few-frontal-and-two-precordial leads, few-frontal-and-one-precordial leads, and lastly, exclusively frontal leads.

### 5.2. Performance Metrics

We evaluate the accuracy of the proposed model in estimating the spatial QRS-T angle with four performance metrics: the absolute mean ($\bar{\epsilon }$) and median ($\tilde{\epsilon }$) estimation errors, the root-mean-squared-error (RMSE), and the Spearman’s rank correlation coefficient ρ between the target α and the estimated $\hat{\alpha }$ angles. The absolute estimation error between an observation *i* is quantified as: $\epsilon_{i}=\left|{\hat{\alpha }}_{i}-\alpha_{i}\right|$, whereas $\bar{\epsilon }$, $\tilde{\epsilon }$, and RMSE as: (13)$$\bar{\epsilon }=\frac{1}{r}\;\sum_{i=1}^{r}\epsilon_{i}\;;\;\;\tilde{\epsilon }=\mathrm{median}\left(\epsilon_{1},\ldots ,\epsilon_{r}\right);\;\;RMSE=\sqrt{\frac{1}{r}\;\sum_{i=1}^{r}{\left({\hat{\alpha }}_{i}-\alpha_{i}\right)}^{2}},$$
and *r* is the total recordings (3873) in the validation dataset. Since the Kolmogorov-Smirnov test shows that the distribution of ϵ is non-normal, $\bar{\epsilon }$ and $\tilde{\epsilon }$ are computed with the nonparametric *bootstrap* method [40] with a resampling of 5000 times. Other metrics shown in the diagrams of Section 6, are also approximated with *bootstrap*: the 95% confidence intervals of $\bar{\epsilon }$ estimated with the bias-corrected percentile method [41]; and the bias and interquartile ranges (iqr) to define the limits-of-agreement in the Bland-Altman plots. The limits-of-agreement are stipulated as 1.45iqr. All the results presented in this article are obtained from the validation dataset.

## 6. Results

Results showing the influence of various hyperparameters on the performance of our proposed model are presented in Section 6.1, whereas Section 6.2 discloses the performance evaluation of the best configuration for each set of ECGs. The presented subset of leads are the subsets with the lowest error $\tilde{\epsilon }$. The recordings in the validation dataset are divided as healthy (class *NORM*) and cardiac disease (classes *MI*, *CD*, *STTC*, *HYP*, and *LOWM*).

### 6.1. Influence of Hyperparameter Tuning on the Model Performance

Figure 7 displays the performance of the proposed deep learning model to estimate α from leads *XYZ* when trained with various combinations of $w_{1}$ and $w_{2}$. An initial number of kernels *k* = 8 suffices to obtain a satisfactory accuracy from leads *XYZ*. Only the depth at which the lowest median error $\tilde{\epsilon }$ was obtained for each combination of $w_{1}$ and $w_{2}$ is shown. Although the lowest $\tilde{\epsilon }$ was reached with {$w_{1}$ = 1.2, $w_{2}$ = 0.2} at *D*= 3 ($\tilde{\epsilon }$ = 3.1°), the model trained with {$w_{1}$ = 0.8, $w_{2}$ = 0.2} at *D*= 4 ($\tilde{\epsilon }$ = 3.3°) achieved the narrowest interquartile range (4.6° vs. 5.1°) and the best overall results throughout all ranges of α. In particular, this configuration outperformed the others for α ≥ 90°, showing lower absolute mean errors $\bar{\epsilon }$ (Figure 7b) despite the smaller number of recordings (samples) in the training dataset for such ranges.

As hypothesized, prioritizing the Euclidean distance ($w_{1}\;L_{d}$) over the QRS-T angle ($w_{2}\;L_{\alpha }$) as the predominant penalization factor, that is, $w_{1}$ > $w_{2}$, results in smaller errors. Combining the Euclidean distance and the angle in the loss function yields better results than using each metric alone ({$w_{1}$ = 1.0, $w_{2}$ = 0.0} and vice versa).

The model trained with the same hyperparameters {$w_{1}$ = 0.8, $w_{2}$ = 0.2} achieved the lowest $\tilde{\epsilon }$ at a smaller depth (*D* = 3) for all investigated sets of reduced-lead ECGs, but required more initial kernels (*k* = 16).

Concatenating metadata (sex and age) resulted in lower $\tilde{\epsilon }$, especially for subsets of reduced-lead ECGs; yet, the improvement in performance was not significant (≤1.5°).

### 6.2. Performance of the Best Configurations on Estimating the Spatial QRS-T Angle

Figure 8, Figure 9 and Figure 10 show in detail the validation performance of the best model configuration in estimating the spatial QRS-T angle using leads *XYZ* and various sets of reduced-lead ECGs: two precodial leads {*I*, *aVF*, *V2*, *V6*}, one precordial lead {*I*, *II*, *aVF*, *V2*}, and solely frontal leads {*I*, *II*, *aVL*, *aVR*}. Table 1 discloses the obtained performance evaluation metrics (*RMSE*, $\tilde{\epsilon }$, and $\bar{\epsilon }$) for each investigated set of leads in the validation dataset.

Figure 8 shows the agreement between the estimated $\hat{\alpha }$ and target α. Even though the estimation errors naturally increase with the reduction of spatial information available in the input leads, the results indicate that reduced-lead estimation of the QRS-T angle is achievable. In the whole validation dataset, the correlation between $\hat{\alpha }$ and α, albeit strong, decreased from ρ = 0.96 for leads *XYZ* that contain all spatial information, to ρ = 0.91 for leads {*I*, *aVF*, *V2*, *V6*} (two precordial), ρ = 0.9 for {*I*, *II*, *aVF*, *V2*} (one precordial), and ρ = 0.77 for {*I*, *II*, *aVL*, *aVR*} (solely frontal).

Despite *RMSE*, $\bar{\epsilon }$, and $\tilde{\epsilon }$ always being higher in ECGs with cardiac disease than the healthy ones, regardless of the subset of leads, $\hat{\alpha }$ and α are more strongly correlated in all morphological classes with cardiac disease than *NORM* for sets of reduced-lead ECGs. The agreement between $\hat{\alpha }$ and α decreases from ρ = 0.86 (*NORM*) vs. ρ = 0.91 (cardiac disease) for {*I*, *aVF*, *V2*, *V6*}; to ρ = 0.85 (*NORM*) vs. ρ = 0.9 (cardiac disease) for {*I*, *II*, *aVF*, *V2*}; and even smaller for {*I*, *II*, *aVL*, *aVF*} with ρ = 0.55 (*NORM*) vs. ρ = 0.81 (cardiac disease). Since $\bar{\epsilon }$ is much lower in the ranges of 5°≤α<70° that are substantially more represented in the training dataset (Figure 9), this correlation decline may be ascribed to higher errors in the underrepresented ranges of α. In leads *XYZ*, ρ = 0.98 for *NORM* recordings, and ρ = 0.95 for cardiac disease.

Figure 9 displays the variation of $\bar{\epsilon }$ in the various sets of leads. The model exhibited markedly higher estimation errors in ranges of α underrepresented in the training dataset (< 200 recordings): α<5° and α≥70° for healthy (*NORM*) ECGs; and α<15° and α≥115° for ECGs with cardiac disease. The downsizing of input precordial leads exacerbated the drop in accuracy, with the set {*I*, *II*, *aVL*, *aVR*} showing the highest susceptibility to estimation errors in the underrepresented ranges of α. The error $\bar{\epsilon }$ is significantly lower in the ranges of α containing more than 200 samples in the training dataset (see Table 1). Nevertheless, $\bar{\epsilon }$ rises as anticipated with the reduction of the spatial information available in the input leads.

Interestingly, in ECGS with cardiac disease, leads *XYZ*, as opposed to any subset of reduced-lead ECGs, displayed the highest estimation errors in ranges of α>115°. Loss of crucial diagnostic information in pathological ECGs caused by the VCG reconstruction method might explain such an unexpected result.

Bland-Altman diagrams in Figure 10 corroborate the abovementioned results. The limits of agreement between $\hat{\alpha }-\alpha$ and α are narrower in leads *XYZ* and start to broaden as the number of precordial leads decreases, with recordings of class *NORM* having less variability from the median bias than those with cardiac disease. In {*I*, *II*, *aVL*, *aVF*}, however, the model reveals an inversely proportional, yet homoscedastic bias, i.e., the variance across different ranges of α is similar. Homoscedasticity is characteristic of models with a variable that has not been fully enclosed. In this case, the sagittal and transverse components that the z-axis supplies. Nevertheless, the inversely proportional bias is not a favorable outcome for cardiovascular health assessment. The model would underestimate the ranges of α>110° associated with an increased risk of dangerous cardiac events.

Figure 11 displays the distribution of the Euclidean distance $L_{d}\left(\vec{u},\hat{\vec{u}}\right)$ between ${\vec{u}}_{QRS}$ and ${\hat{\vec{u}}}_{QRS}$, and ${\vec{u}}_{T}$ and ${\hat{\vec{u}}}_{T}$ in each plane: XY (frontal), XZ (transverse), and YZ (sagittal). The distance is calculated as the projection of $\vec{u}$ and $\hat{\vec{u}}$ in the respective plane. $L_{d}\left(\vec{u},\hat{\vec{u}}\right)$ gradually lengthens in every plane from leads *XYZ* to {*I*, *aVF*, *V2*, *V6*} and {*I*, *II*, *aVF*, *V2*} but becomes discernibly higher in the XZ and YZ planes in frontal leads {*I*, *II*, *aVL*, *aVF*}, which only carry information in the XY plane. Larger $L_{d}\left(\vec{u},\hat{\vec{u}}\right)$ suggests the model encountered extra obstacles to locate the vector’s coordinates within the specified plane.

## 7. Discussion

### 7.1. Summary and Significance

Monitoring the spatial QRS-T angle, evidenced as one of the most propitious markers for risk assessment of SCD [8,9], was presumed to be impracticable in out-of-hospital settings thus far. Our research introduces a deep-learning-based method to measure the spatial QRS-T angle using a set of reduced-lead ECGs that can conveniently be recorded with consumer healthcare devices. Our proposed model, albeit prototypal, sparks scientific interest in engineering methods for ambulatory monitoring of the spatial QRS-T angle, which can lead to substantial contributions toward harnessing the diagnostic value of QRS-T angle for cardiovascular health assessment in free-living conditions. To the best of our knowledge, this is the first study to examine whether it is conceivable to estimate the spatial QRS-T angle from reduced-lead ECGs.

### 7.2. Considerations on the Model Architecture

The baseline architecture of our model is engineered to be accurate yet simple enough to be lightweight and have the low computational power to be integrated into consumer healthcare devices. Compared to other CNN1Ds for ECG analysis, often comprised of 8-to-34 [16,18,20,42] blocks of layers, our baseline architecture of three-to-four blocks (*D* = {3, 4}) and *k* = 16 suffices to get satisfactory results. Our proposed model contains only 105,578 trainable parameters, nearly 12 times less than the CNN1D developed for the classification of single-lead ambulatory ECGs [18]. While popular due to their high accuracy, deeper neural networks also entail larger training datasets and computational resources that can hamper the deployment of the network in devices with hardware and computational constraints such as wearables. Furthermore, adopting deeper neural networks does not necessarily translate into significant improvements in accuracy to justify the tradeoffs in resources if the goal application is for out-of-hospital monitoring of the QRS-T angle.

Smaller networks like ours, or as in the one applied for automatic diagnosis of 12-lead ECGs [42], outperform their convolutional-blocks-only counterparts when enhanced with custom blocks such as residual connections, squeeze-and-excitation, atrous spatial pooling, or case-specific loss functions [32,33]. Our strategy involved residual blocks with a predominant focus on an original loss function. Our proposed loss function combines two metrics, each with their penalization weight, to train the model: the Euclidean distance ($w_{1}$) and the QRS-T angle ($w_{2}$). Prioritizing the Euclidean distance over the QRS-T angle (i.e., $w_{1}$ > $w_{2}$) as the main penalization factor results in smaller errors, namely in sets of reduced-leads ECGs. Optimization with the Euclidean distance combined with the QRS-T angle instead of the QRS-T angle alone allows the model to recognize that ECGs with visible differences in morphology can still have similar QRS-T angles, minimizing the chances of the model associating a specific morphology to a particular range of α. Morphologically different ECGs with similar QRS-T angles are often the case in patients with distinctive cardiac conduction axes in which the direction of the overall electrical activity of the heart is not the same. In a 3D space, this means that the vectors ${\vec{u}}_{QRS}$ and ${\vec{u}}_{T}$ of each patient are located in different planes (octants), but the angle between them does not necessarily differ. Searching for the coordinates of both target vectors helps the model leverage any available information to boost accuracy. Thus, adopting metrics that guide the model in the 3D space is a favorable choice.

### 7.3. Considerations on the Attained Results

The model demonstrated a propensity to higher estimation errors in ranges of QRS-T angle represented by less than 200 recordings (samples) per morphological class in the training data. This propensity is amplified as the number of input precordial leads decreases. Although an increase in estimation errors is anticipated with the reduction of spatial information available in the input leads, the interconnection between higher errors and fewer training samples suggests that additional recordings may promote a more accurate QRS-T angle estimation from reduced-lead ECGs. When the complete spatial information is accessible in leads *XYZ*, the model can straightforwardly identify relevant data features from fewer recordings. Conversely, the relevant data features may be less conspicuous and more challenging to detect in reduced-lead ECGs with limited spatial information. Thus, the model may necessitate more training samples to identify relevant data features.

Surprisingly, leads *XYZ*, but not subsets of reduced-lead ECGs, showed the highest estimation errors in ECGs with cardiac disease in the underrepresented ranges of α≥ 115°. Such an unexpected result may be attributed to possible signal distortions caused during the VCG reconstruction process. In certain cardiac pathologies, the mathematical transformations to derive the VCG can camouflage (or even eliminate) distinctive data features [14,43], hindering the model’s ability to recognize any feature patterns that point to the location of ${\vec{u}}_{QRS}$ and ${\vec{u}}_{T}$. In contrast, these distinctive data features are preserved in ECGs of cardiac pathology, even in frontal leads, hence impelling the model to locate the target vectors more correctly.

An analogous argument can also explain the accuracy drop in the estimated $\hat{\alpha }$ from any subset of reduced-lead ECGs of class *NORM*, in which the estimation errors were substantially higher in the underrepresented ranges of α≥70°. Since wider QRS-T angles are generally associated with severe cardiac diseases [9,44], such a surprising result raises the question of whether large values of α can occur in healthy ECGs or are ascribed to label noise. However, label noise could only justify such a result if leads *XYZ* displayed the same discrepancy in estimation errors in the ranges of α observed in reduced-lead ECGs. A plausible explanation lies in the sagittal (YZ) plane, which may contain the most indicative data features of wider QRS-T angles in the absence of cardiac disease. With only fragments of sagittal information given in reduced-lead ECGs, the model struggles to identify data features characteristic of wide QRS-T angles in seemingly healthy ECGs if fewer training samples are provided.

Correctly estimating the location of ${\vec{u}}_{QRS}$ and ${\vec{u}}_{T}$ in any plane incorporating the orthogonal lead *Z* (sagittal and transverse) is challenging in reduced-lead ECGs regardless of the morphological class. Reducing the amount of spatial information in the input ECG-leads encumbers the search for the coordinates of ${\vec{u}}_{QRS}$ and ${\vec{u}}_{T}$, namely through the z-axis, as verified by an increase of the Euclidean distance between the target and estimated vectors. In parallel with additional recordings, decomposing the Euclidean distance loss into each of the three planes could be a potential solution to enhance the location accuracy of the z-coordinate. Isolating the planes in the loss function enables tailoring of the penalization factor to each plane, which may promote better estimation results.

### 7.4. Suitability for ECG Consumer Healthcare Devices

While our investigation consisted only of non-ambulatory ECGs collected with clinical devices, the suitability of the proposed model for consumer healthcare devices is plausible and merits further discussion. Conceptually, ambulatory estimation of the spatial QRS-T angle can be performed similarly to KardiaMobile® [45] for arrhythmia detection or the prototype technology developed to monitor electrolyte fluctuations in hemodialysis patients at home [46]. Pre-processing, heartbeat averaging and subsequent QRS-T angle computation are feasible offline with some delay for short intermittent ECG recordings (15-to-60 s) or through cloud processing for longer recordings. Cloud processing would also support the transmission of estimated QRS-T angles to health professionals for remote verification of potentially dangerous cardiac events.

An attractive attribute of the proposed deep learning model is its simplicity. When looking at the computational demands of the whole algorithm, the QRS-T angle can be estimated swiftly, with the preprocessing stage exercising more computational time and resources than the deep learning model itself. In recordings scenarios that assure that 10-to-15 s long ECGs are registered with sufficient quality to warrant low-complexity filtering in the preprocessing stage, the spatial QRS-T angle can be calculated in a few seconds with the advantage of not needing PQRST delineation, which is often problematic in ambulatory recordings due to noise.

Our model measured the spatial QRS-T angle with reasonable accuracy from a set of three frontal-and-one precordial leads ({*I*, *II*, *aVF*, *V2*}) that can be registered with three electrodes instead of the eight required to derive the QRS-T angle using the conventional approach. Requiring one precordial lead evidently restricts the type of consumer healthcare devices suitable for deploying our deep learning model in future applications, precluding the use of devices that maximize comfort, such as wrist-worn wearables [47], which only register frontal-lead ECGs. Nevertheless, the market already offers a handful of practical devices that acquire frontal-and-one precordial lead ECGs with an acceptable degree of comfortableness [48], namely those patch-based (e.g., Bittium OmegaSnap™ [49]) or contact-based textile (e.g., Viscero ECG vest [50]) ECG electrodes. A downsize of eight to three electrodes is still a substantive improvement. Even if the comfort level of three electrodes is lower than that of other wearables, the existing patch- or textile-based ECG devices are durable, easy to configure, and may be adequate for intermittent monitoring of the QRS-T angle in out-of-hospital settings. Recent advancements in the reconstruction of the standard 12-lead ECG from sets of reduced-lead ECGs have, however, demonstrated to be possible to derive lead *V2* from lead *II* [51] in healthy subjects. The encouraging preliminary results indicate an appealing solution for estimating the spatial QRS-T angle with comfortable wearable devices in the future.

Most commercialized ECG consumer healthcare devices have technical specifications analogous to the clinical recordings used in our research: (i) a minimum 16-bit precision at a resolution of 1 μV/LSB, (ii) ECG bandwidth of 0.05 Hz to at least 100 Hz, and (iii) sampling rates starting at 200 Hz. However, since the estimation of the spatial QRS-T angles has been performed exclusively from clinical devices with higher signal resolution and sampling rates to date, the minimal technical specifications of ECG consumer healthcare devices suitable for ambulatory measurement remain undefined. While our model estimated QRS-T angles from heartbeats downsampled to 250 Hz with satisfactory accuracy, we did not investigate the influence of different technical specifications on the estimation of the QRS-T angle, nor we are aware of studies that examined this question. Although higher ECG bandwidths and sampling rates ≥ 500 Hz are pertinent for detecting arrhythmias [38] and pediatric ECGs [52] and are often recommended for clinical ECGs [38], deep learning models can detect arrhythmias from ECGs with sampling rates of 300 Hz [17], 200 Hz [18], and even 100 Hz [19]. Thus, a minimum sampling rate of 200 Hz seems a reasonable compromise between adequate deep-learning-based analysis of ambulatory ECGs without increasing hardware and computational complexity or draining the battery life of consumer healthcare devices [53].

### 7.5. Limitations and Future Directions

When considering the ultimate application goal of our research, which is to facilitate ambulatory monitoring of the spatial QRS-T angle for cardiovascular health assessment, one must pose two central questions: (i) What is the maximum acceptable estimation error; and most importantly, (ii) What is the clinical value of an 1° increase. All medical research regarding the prognostic value of the QRS-T angle focuses on observational studies [8,9] with follow-up periods of 2-to-30 years that categorize the angle into subranges, most commonly as normal (<110°) or dangerous (≥110°) [9,54]. While the optimal cut-off threshold for assessing the dangerousness level of the spatial QRS-T angle depends on sex [30], age [31], medical history [8], and even the methods to derive the ${\vec{u}}_{QRS}$ and ${\vec{u}}_{T}$ [27,43], no studies investigated if day-to-day fluctuations of the QRS-T angle offer any clinical value. Populations susceptible to abnormalities in cardiac repolarization, such as chronic kidney disease patients, could benefit from daily monitoring of QRS-T angle fluctuations, in which the angle variation would be more auspicious than the absolute value itself. Higher estimation errors may be acceptable for such application scenarios, providing that the bias is constant. Oppositely, scenarios aiming to classify subranges of spatial QRS-T angles per clinical importance may require smaller estimation errors within the predefined cut-off ranges.

Unfortunately, the answer to the posed questions remains open and falls beyond the scope of our research. Nevertheless, considering that intra-subject inaccuracies up to 10° are to be expected [27] and that a 20° increase in the spatial QRS-T angle is associated with a 4% aggravated risk of mortality [55], the estimation errors obtained from the subset {*I*, *II*, *aVF*, *V2*} may suffice for detecting abnormal QRS-T angles without compromising patient comfortability. Furthermore, measuring the spatial QRS-T angle from subsets of solely frontal leads looks plausible in the future with further refinements of our deep learning prototype model.

A viable solution for boosting the estimation of spatial QRS-T angle from reduced-lead ECGs could be the adoption of ensemble methods in hierarchical order. Ensemble methods could first classify ECGs into various subranges (classes) of QRS-T angles and subsequently assign different regression networks to each separate class. The designated classes could categorize ranges of spatial QRS-T angles by clinical significance: narrow (0°≤α<30°), healthy (30°≤α<70°), borderline healthy (70°≤α<110°), dangerous (110°≤α<135°), life-threatening (135°≤α≤180°). This strategy would narrow the scope of the angle to be measured, enabling the selection of regression networks (or loss functions) better fitted to handle specific challenges within each subrange of QRS-T angles. For instance, if the model struggles to locate the z-coordinate only in ranges of α≥70°, a higher penalization factor of the Euclidean distance or a deeper regression network could be appropriate options to train the model for such ranges of the QRS-T angle. Alternatively, other subranges of QRS-T angles could benefit from different regression algorithms such as ElasticNet. Cascading the estimation of the QRS-T angle from reduced-lead ECGS is also a compelling solution to mitigate the shortage of training data in given ranges of QRS-T angle. In particular, in small (α < 15°) or wide (α > 135°) QRS-T angles. Accurate measurement of the actual value of the QRS-T angle from reduced-lead ECGS for such ranges may be unattainable if the number of recordings for training the model is low, but grouping the recordings into different classes may yield satisfactory classification results. Considering that spatial QRS-T angles starting from α = [110:135]° are associated with an elevated risk of SCD [9], correctly categorizing the ECG as a life-threatening QRS-T angle (α > 135°) would provide sufficient clinical value.

The lack of ambulatory long-term ECG signals is a limitation of our research. The *PTB-XL* database, albeit comprehensive in terms of healthy controls and cardiac pathologies, includes only short 10 s long ECGs; therefore, it is unclear how will noisy real-life ECG signals impact the performance of the deep-learning-based method for spatial QRS-T angle estimation. The frontal QRS-T angle is also useful in predicting mortality [9] and ventricular arrhythmias [56]. Future studies should also investigate monitoring of the frontal, along with the spatial, QRS-T angle from a reduced ECG lead system.

Future research should also consider supplementing their training data with additional recordings, namely of the underrepresented ranges of α< 15° and α≥ 115°, either by data augmentation techniques or inclusion of other datasets of 12-lead ECGs likely to display α≥ 115° to complement the *PTB-XL* dataset.

## 8. Conclusions

Estimation of spatial QRS-T angle from reduced-lead ECGs is indeed conceivable. Our proposed deep-learning-based method estimated the spatial QRS-T angle from few-frontal-and-one precordial leads {*I*, *II*, *aVF*, *V2*} with sufficient accuracy for detecting abnormal QRS-T angles without sacrificing patient comfortability. The reduced set of ECG leads can be conveniently registered with easy-to-use patch- or textile-based consumer healthcare devices already available in the market. In contrast to other CNN1Ds for ambulatory ECG analysis, the architecture of our deep learning model is smaller and more lightweight, which is preferable for deployment in consumer healthcare devices. The designed architecture incorporates custom blocks like residual connections and an original composite loss function to boost the accuracy of the estimated QRS-T angles without increments in computational complexity. The proposed model was trained and validated by gradually reducing the ECG leads from a publicly available dataset of clinical 12-lead ECG recordings. Our research can catalyze scientific interest in developing methods to estimate the QRS-T angle from reduced-lead ECGs. Such methods, like the proposed deep learning model, are a step toward facilitating prolonged monitoring of the spatial QRS-T angle for cardiovascular health assessment in free-living conditions. Future research can validate the potential clinical benefits of this technology in populations at risk of dangerous cardiac events, such as patients with chronic cardiac or kidney disease.

## Figures and Tables

**Figure 1 sensors-22-05414-f001:**
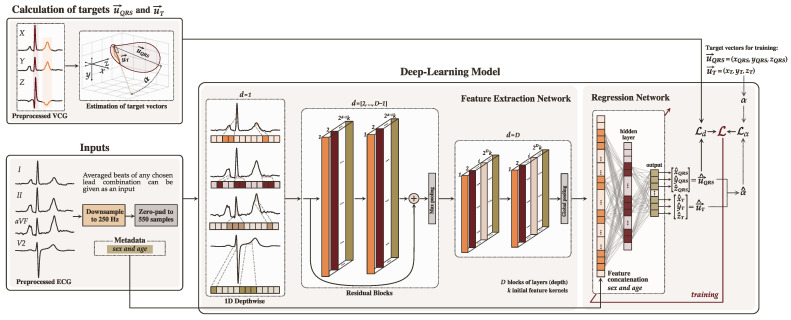
Overview of the proposed deep learning model for estimation of QRS-T angle using reduced-lead ECGs. The model is composed of two parts: feature extraction and regression. The target vectors ${\vec{u}}_{QRS}$ and ${\vec{u}}_{T}$ and spatial QRS-T angle α are computed from VCGs.

**Figure 2 sensors-22-05414-f002:**
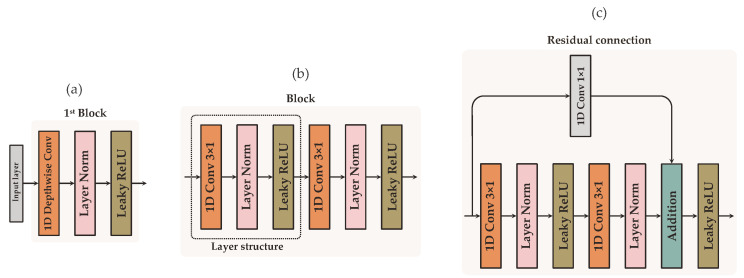
Detailed representation of the three types of blocks employed in the feature extraction network: (**a**) first block, (**b**) the last block, and (**c**) blocks with residual connections.

**Figure 3 sensors-22-05414-f003:**
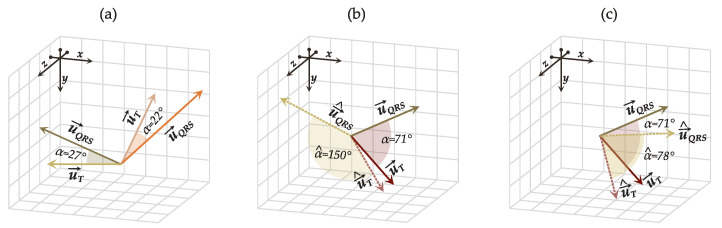
Case scenarios of: (**a**) similar QRS-T angles α of two ${\vec{u}}_{QRS}$ and ${\vec{u}}_{T}$ located in two different planes; (**b**) correct location of one vector (${\vec{u}}_{T}$) but not the other (${\vec{u}}_{QRS}$), yielding large errors in the estimated QRS-T angle $\hat{\alpha }$; (**c**) compromise between minor errors in the location of both ${\vec{u}}_{QRS}$ and ${\vec{u}}_{T}$ to achieve a more accurate QRS-T angle estimation.

**Figure 4 sensors-22-05414-f004:**
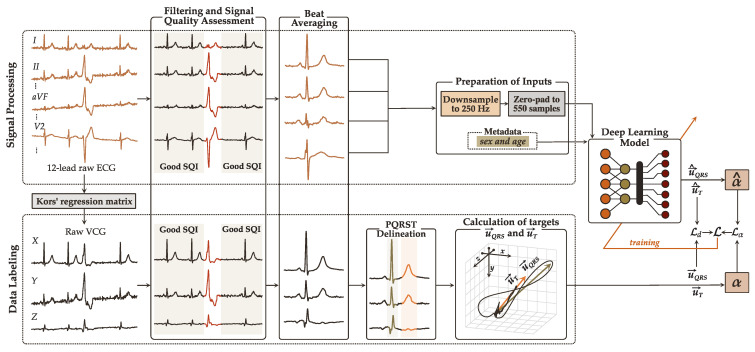
Data preparation and labeling. Signals undergo preprocessing to generate the input signal-averaged beats.

**Figure 5 sensors-22-05414-f005:**
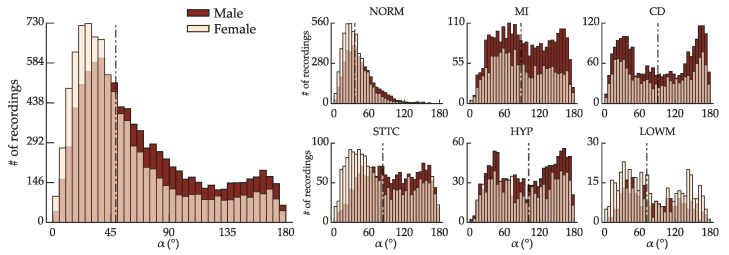
Distribution of spatial QRS-T angles α of across the ranges of α = [0:5:180]° according to sex (overlapped) for all eligible recordings in the dataset (**left**) and for each morphological class (**right**). α is the angle between the VCG vectors ${\vec{u}}_{QRS}$ and ${\vec{u}}_{T}$. The dashed line is the median α for each class.

**Figure 6 sensors-22-05414-f006:**
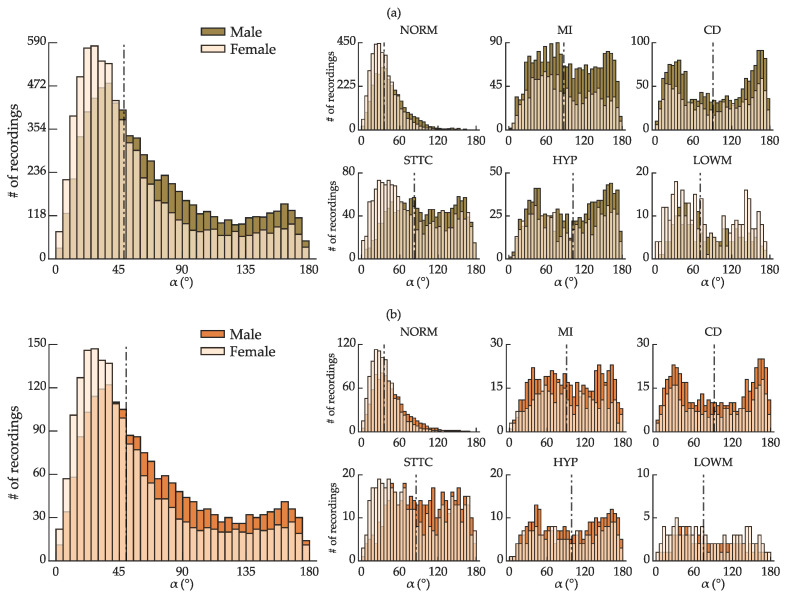
Distribution of spatial QRS-T angle α across the ranges of α = [0:5:180]° according to sex (overlapped) for all recordings suitable for analysis (**left**) and for each morphological class (**right**) in the (**a**) training and (**b**) validation sets. The dashed line is the median α for each class.

**Figure 7 sensors-22-05414-f007:**
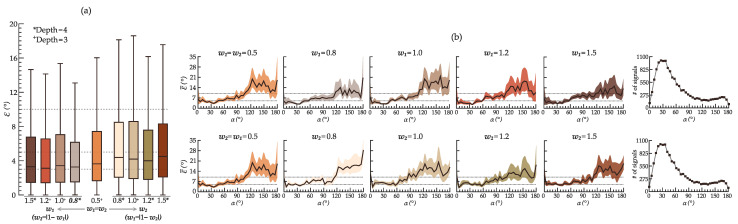
Performance of various model configurations tuned with different combinations of hyperparameters $w_{1}$ and $w_{2}$ in estimating the spatial QRS-T angle from leads *XYZ* in the validation dataset. (**a**) Boxplot of the obtained absolute error ϵ (outliers not shown) for every combination of $w_{1}$ and $w_{2}$. $w_{1}$ increases to the left side, wheres $w_{2}$ to the right. The other hyperparameter value is obtained as |1−w| on each side. (**b**) Mean absolute error $\bar{\epsilon }$ and the respective 95% confidence interval across the ranges of α = [0:5:180]° for increasing $w_{1}$ (**top row**) and $w_{2}$ (**bottom row**). The last column displays the number of training samples for each range of α.

**Figure 8 sensors-22-05414-f008:**
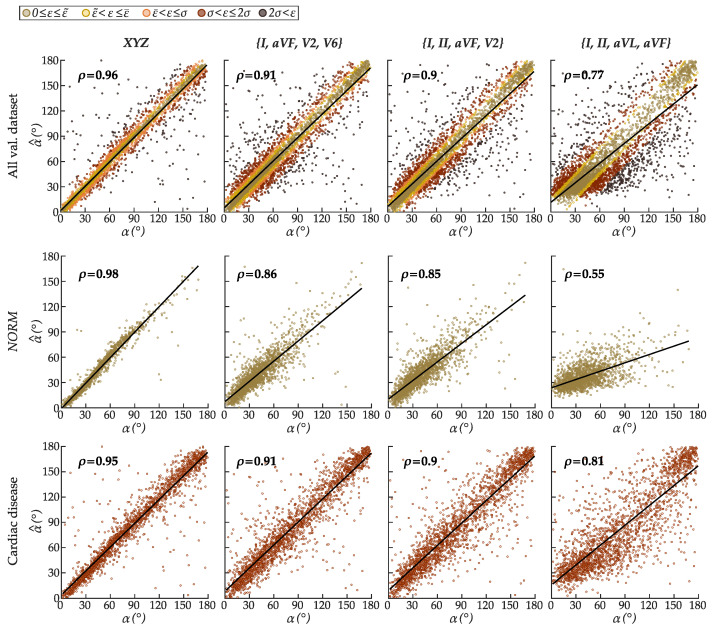
Scatter plot diagrams of the deep-learning-estimated $\hat{\alpha }$ vs. target α from various sets of leads for (**top row**) all recordings, and ECGs with normal (**middle row**) *NORM* and (**bottom row**) cardiac disease in the validation dataset. The estimation error ϵ of every $\hat{\alpha }$ in the first row is color-grouped based on the absolute median ($\tilde{\epsilon }$), mean ($\bar{\epsilon }$), and standard deviation ($\sigma_{\epsilon }$) error.

**Figure 9 sensors-22-05414-f009:**
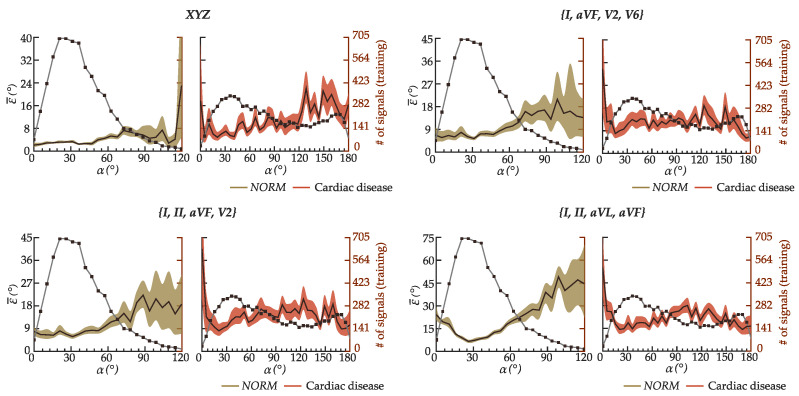
Variation of the mean absolute error $\bar{\epsilon }$ and the respective the 95% confidence interval across the ranges of α = [0:5:180]° for ECGs with normal (*NORM*) and diseased cardiac function. The right axis indicates the number of training samples in each range of α. Since the number of *NORM* subjects with α > 120° is almost negligible, $\bar{\epsilon }$ is not shown for these ranges of α.

**Figure 10 sensors-22-05414-f010:**
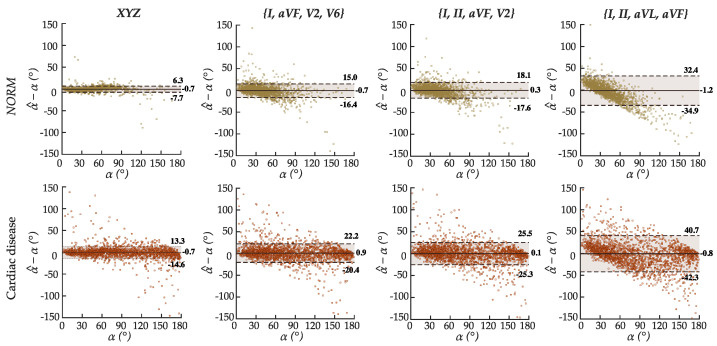
Bland-Altman diagrams of deep-learning-based estimation of $\hat{\alpha }$ from various sets of leads of ECGs with (**top**) normal (*NORM*) and (**bottom**) diseased cardiac function.

**Figure 11 sensors-22-05414-f011:**
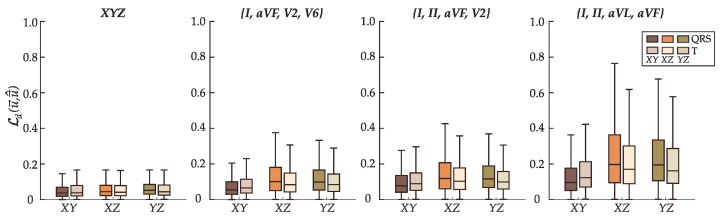
Distribution of the Euclidean distance $L_{d}\left(\vec{u},\hat{\vec{u}}\right)$ between ${\vec{u}}_{QRS}$ and ${\hat{\vec{u}}}_{QRS}$, and ${\vec{u}}_{T}$ and ${\hat{\vec{u}}}_{T}$ in each of the three planes: XY (frontal), XZ (transverse), and YZ (sagittal).

**Table 1 sensors-22-05414-t001:** Performance deep-learning-based estimation of the spatial QRS-T angle α from various sets of leads in whole the validation dataset, and in ECGs with healthy and diseased cardiac function.

		**Subset of Leads**											
		* **XYZ** *			**{*I, aVF, V2, V6*}**			**{*I, II, aVF, V2*}**			**{*I, II, aVL, aVR*}**		
**Recordings**	**Ranges of α**	* **RMSE** *	$\bar{\epsilon }$	$\tilde{\epsilon }$	* **RMSE** *	$\bar{\epsilon }$	$\tilde{\epsilon }$	* **RMSE** *	$\bar{\epsilon }$	$\tilde{\epsilon }$	* **RMSE** *	$\bar{\epsilon }$	$\tilde{\epsilon }$
All val. dataset	0°≤α≤180°	12.2°	5.8°	3.3°	17.2°	10.3°	6.4°	18.4°	11.4°	7.3°	25.4°	17.9°	12.7°
	5°≤α<115° ${}^{1}$	9.2°	4.7°	2.9°	15.4°	9.8°	6.3°	16.0°	10.5°	7.1°	22.8°	16.6°	12.2°
*NORM*	0°≤α≤180°	6.1°	3.4°	2.5°	13.5°	8.3°	5.5°	14.1°	9.0°	6.1°	21.0°	14.9°	11.1°
	5°≤α<70° ${}^{1}$	4.6°	3.0°	2.4°	11.0°	7.2°	5.1°	11.1°	7.6°	5.7°	15.2°	11.7°	9.8°
Cardiac disease	0°≤α≤180°	16.8°	8.7°	4.9°	20.5°	12.2°	7.3°	21.8°	13.7°	8.7°	29.1°	20.7°	14.3°
	15°≤α<115° ${}^{1}$	12.8°	7.2°	4.2°	18.4°	12.1°	8.1°	19.6°	13.3°	9.5°	27.7°	20.6°	15.0°

^1^ Ranges of *α* adequately represented in the training dataset (>200 samples).

## Data Availability

The *PTB-XL* dataset employed in this research is freely accessible in PhysioNet at https://physionet.org/content/ptb-xl/1.0.1/.

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
