# Peer review of "Deep-Learning-Based Estimation of the Spatial QRS-T Angle from Reduced-Lead ECGs"

_sensors, 2022, doi:10.3390/s22145414_

Round 1
Reviewer 1 Report
The authors present a very neat research study, a 1D convolutional neural network to estimate the spatial QRS-T angle from signal-averaged heartbeats of reduced-lead ECGs. I suggest the authors consider answering the following questions so that the audience can better appreciate the work and since ambulatory recordings are what your model will end up using as inputs.
-Provide a short relator of which commercialized consumer healthcare devices were investigated? What are the characteristics of those consumer healthcare devices? About the software for measuring, the common specs or any special tool. Since you mention to have investigated many of them but you only reference a few briefly.
-Do you know of any ambulatory ECG monitoring standard that promotes QRS-T angle measurement? Is there a best signal resolution or sample rate spec for this measurement? It seems from your document that there is no issue related to resolution and sampling rate for the for measuring the QRS-T angle.
-Are you proposing to integrate your neural network model into an ambulatory device? Why might this be a market need? Can your model also be used offline or through cloud processing?
-How does signal-averaged heartbeat work in an ambulatory device? Is there any consumer healthcare device that does it? I understood that this processing is necessary as the input to your model
-What is the idea for recordings upsampled to 500 Hz, then downsampling to 250 Hz to then zero-pad to 550 samples? Can You explain this better?
-"the architecture of our deep learning model is lightweight enough to be deployed in consumer healthcare devices", please provide a reference for this to be valid.
-What was the R-peak detection algorithm used? If any
Reviewer 2 Report
In the article, the authors propose a new approach for the QRS-T spatial angle estimating base on a smaller number of ECG leads. The assessment is based on deep machine learning methods. To confirm the main ideas of the work, the authors use the PTB-XL dataset.
The paper includes all necessary procedures for effective classification into one of the given classes. In particular, filtering of ECG signals, alignment of the baseline, exclusion of some cardiograms, which may affect the reliability of recognition, is carried out.
The problems and difficulties of finding the QRS-T angle and methods of overcoming them are described in sufficient detail.
The main result of the paper is presented in section 3, where a deep learning model is proposed for estimating the QRS-T spatial angle using a set of ECGs taken from a smaller number of leads.
Remarks:
1. In paragraph 4.1.1. it is indicated "Recordings with at least one lead that contains more than 50% poor-quality beats within the 10 s ECG are considered unanalyzable and hence discarded. ECGs with discernible rhythm disturbances, such as atrial or ventricular flutter or fibrillation, are also excluded from the analysis...", lines 216-219. But it is not specified from which ECGs it is rejected. This makes it impossible for other scientists to verify the correctness of the results obtained in this paper.
2. The results of Chapter 6 are described in great detail with many technical parameters that are difficult to understand. Figures 7-11 have a lot of 11 graphs each, which are difficult to distinguish and understand.
3. The article is written on a very important and modern topic, but the list of references is very outdated. Perhaps you need to make a reference to more modern sources from the last 5 years.
Reviewer 3 Report
The manuscript presents a deep learning based methodology for measurement of the spatial QRS-T angle as a marker for the risk of sudden cardiac death (SCD) using a reduced ECG lead sets. The authors show performance results for different lead sets declaring sufficient accuracy for the lead set {I,II,aVF,V2}, which is suitable for different wearable ECG devices.
The manuscript is well written, easy for reading and understanding. It would be suitable for publication after some minor corrections as follows:
1) In the first row of the abstract – correct “QTS-T” to “QRS-T”.
2) In the block-diagram in Figure 1 and also in the text the authors have written in the Residual block ‘d’ changes from 2 to D-2. I suppose d=[2, …, D-1].
3) Figure 4 – are X,Y,Z downsampled before the delineation? From the block-diagram it seems that they are not downsampled before the delineation.
4) Table 1 – the correspondence between ‘Class’ (3 classes) and ‘Range of alpha’ (4 ranges) is not clear. How do the values in the table (presented in 4 rows) correspond to the 3 classes in the first column? Authors refer to alpha value of 115 degrees in the text, however, such value (limit) is not present in the table.
